# CD8/PD-L1 immunohistochemical reactivity and gene alterations in cutaneous squamous cell carcinoma

**Haruto Nishida** ⬡*, **Yoshihiko Kondo, Takahiro Kusaba, Kazuhiro Kawamura, Yuzo Oyama, Tsutomu Daa**

Department of Diagnostic Pathology, Faculty of Medicine, Oita University, Oita, Japan

* nharuto@oita-u.ac.jp

**Data Availability Statement:** All relevant data are within the paper.

## Abstract

In recent years, several immune checkpoint inhibitors targeting programmed death-ligand 1 (PD-L1) or PD-1 have been developed for cancer therapy. The genetic background of tumors and factors that influence PD-L1 expression in tumor tissues are not yet elucidated in cutaneous squamous cell carcinoma (cSCC). CD8-positive tumor-infiltrating lymphocytes (TILs) are known to be related to tumor immunity. Here, we aimed to study the relationship between CD8/PD-L1 immunohistochemical reactivity and gene alterations in cSCC. Tumorigenic genes were examined to identify gene alterations using next-generation sequencing (NGS). We collected 27 cSCC tissue samples (from 13 metastatic and 14 non-metastatic patients at primary diagnosis). We performed immunohistochemical staining for CD8 and PD-L1, and NGS using a commercially available sequencing panel (Illumina Cancer Hotspot Panel V2) that targets 50 cancer-associated genes. Immunohistochemically, CD8-positive TILs showed a high positive score in cSCC without metastasis; in these cases, cSCC occurred predominantly in sun-exposed areas, the tumor size was smaller, and the total gene variation numbers were notably low. The tumor depth, PD-L1 positivity, and gene variation number with or without tumor metastasis were not related, but the gene variation number tended to be higher in cSCCs arising in non-sun-exposed areas. Tumor metastasis was more common in cSCC arising in non-sun-exposed areas, which decreased the number of TILs or CD8-positive cells. From a genetic perspective, the total gene alterations were higher in cSCC with metastasis. Among them, *ERBB4* and *NPM1* are presumably involved in cSCC tumorigenesis; in addition, *GNAQ*, *GNAS*, *JAK2*, *NRAS*, *IDH2*, and *CTNNB1* may be related to tumor metastasis. These results provide information on potential genes that can be targeted for cSCC therapy and on immune checkpoint inhibitors that may be used for cSCC therapy.

## Introduction

Recently, remarkable progress has been made in the application of immunotherapy as a major strategy for cancer treatment, and several new immunotherapy drugs are now being released

**Funding:** Grants: This study was supported by the KAKENHI grant from the Japan Society for the Promotion of Science KAKENHI Grant (Number 20K16195). The funders had no role in study design, data collection and analysis, decision to publish, or preparation of the manuscript.

**Competing interests:** The authors have declared that no competing interests exist.

[1–3]. In particular, immunotherapy using the anti-programmed death-ligand 1 (PD-L1) antibody, an immune checkpoint inhibitor, is used for cutaneous malignant melanoma and squamous cell carcinoma of the lung and head, and neck [1]. Recently, an anti-PD-1 antibody product, Libtayo, was approved for use in the treatment of cutaneous squamous cell carcinoma (cSCC). Most skin malignancies are squamous cell carcinomas, with good prognoses, but the prognoses of advanced-stage cases are poor [4–6]. A few studies have focused on the genetic alterations in high-risk cSCC, and alterations in the *TP53*, *NOTCH*, and *RAS* families have been reported [5]. Sun exposure leads to mutations in *NOTCH*, *TP53*, *FAT1*, *FGFR3*, and *EGFR* [7]. CD8-positive tumor-infiltrating lymphocytes (TILs) are known to be related to tumor immunity; however, the prognostic factors of cSCC are still unknown. Hence, we aimed to study the relationship between the immunohistochemical reactivity for CD8 and PD-L1 and gene alterations in cSCC. Tumorigenic genes were identified using next-generation sequencing (NGS). This may lead to new therapeutic uses (drug repositioning) and development of new drugs for cSCC.

## Materials and methods

We collected 27 surgically resected cSCC tissue samples (from 13 metastatic and 14 non-metastatic cases at primary diagnosis) between January 2000 and December 2020 at the Oita University Hospital. Formalin-fixed paraffin-embedded tissue (FFPE) blocks that showed representative histology were selected and cut into 4 μm-thick slices for immunohistochemistry analysis or into 10 μm-thick slices for gene sequencing. The study adhered to the guidelines described in the Declaration of Helsinki and was approved by the institutional ethics committee and review board of Oita University, Japan (approval number: 2191). The study was retrospective, and patients' consent was obtained using an opt-out method.

### Immunohistochemistry

Immunohistochemical staining was performed by briefly deparaffinizing the sections in xylene and rehydrating them in a graded series of alcohol; endogenous peroxidase activity was then blocked by incubation with 3% hydrogen peroxide for 20 min at 25˚C. The antigens were retrieved by autoclaving the samples in citrate buffer (pH 9.0), and the slides were incubated with an anti-CD8 antibody (1:50, clone DK25, DAKO, Denmark, Code No. PB 984) for 2 h at 25˚C, retrieved by autoclaving in citrate buffer (pH 6.0), and incubated with PD-L1 antibody (1:30, polyclonal, Abcam, Cambridge, UK, ab233482) for 30 min at 25˚C. Immunoreactions were visualized using a streptavidin-labeled biotin peroxidase complex system (Nichirei, Tokyo, Japan). The nuclei were counterstained with hematoxylin. For evaluating TILs, a focus region in the tumor invasion area of the largest represented specimen was scored as 0 or 1 depending on the absence or presence of CD8-positive cells, respectively (Fig 1) [1]. In the assessment of PD-L1 positivity, the case was scored as 1 when positive staining was detected in >5% of the tumor cells and as 0 otherwise (Fig 2) [1].

### NGS

DNA was extracted from the previously cut 10 μm-thick FFPE sections using a QIAamp DNA FFPE Tissue Kit (Qiagen, Germantown, MD, USA). DNA was quantified using a Qubit 3.0 fluorometer (Life Technologies-Thermo Fisher Scientific, Saint Aubin, France); we used the sample within the standard value according to the manual. Sequencing libraries were prepared for iSeq 100 according to the manual, and AmpliSeq was used for the Illumina Cancer Hotspot Panel V2 (Illumina, San Diego, California, USA) [8, 9]. The panel targeted 50 cancer-associated genes: *ABL1*, *EGFR*, *GNAS*, *KRAS*, *PTPN11*, *AKT1*, *ERBB2*, *GNAQ*, *MET*, *RB1*, *ALK*,

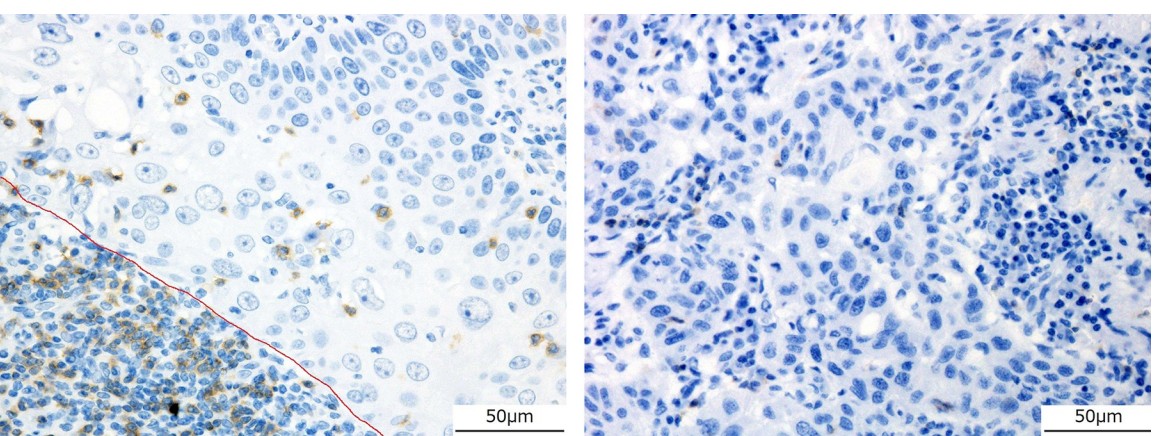

**Fig 1. Immunohistochemical analysis of CD8 (×40).** Immunohistochemical analysis indicating that CD8-positive cells (brown) surrounded and invaded tumor nests. Images of sections with staining scored as 1 (a) and 0 (b). The nuclei were counterstained with hematoxylin.

*ERBB4, HNF1A, MLH1, RET, APC, EZH2, HRAS, MPL, SMAD4, ATM, FBXW7, IDH1, NOTCH1, SMARCB1, BRAF, FGFR1, JAK2, NPM1, SMO, CDH1, FGFR2, JAK3, NRAS, SRC, CDKN2A, FGFR3, IDH2, PDGFRA, STK11, CSF1R, FLT3, KDR, PIK3CA, TP53, CTNNB1, GNA11, KIT, PTEN,* and *VHL*. Mutations were studied using the Illumina VariantStudio software (Illumina). The gene alterations were referred to as stipulated by the gnomAD (https://gnomad.broadinstitute.org/), dbSNP (https://www.ncbi.nlm.nih.gov/snp/), and COSMIC (https://cancer.sanger.ac.uk/cosmic) databases. Gene pathogenicity was predicted using the ClinVar (https://www.ncbi.nlm.nih.gov/clinvar/) and PolyPhen (http://genetics.bwh.harvard.edu/pph2/) databases (in September 2022).

## Statistical analyses

We performed a Chi-square test (for comparing TILs or PD-L1 between SCC without and with metastasis) and a Student's *t*-test (for comparing age, sex, size, tumor depth, and position between SCC without and with metastasis). A *p*-value < 0.05 was considered statistically significant.

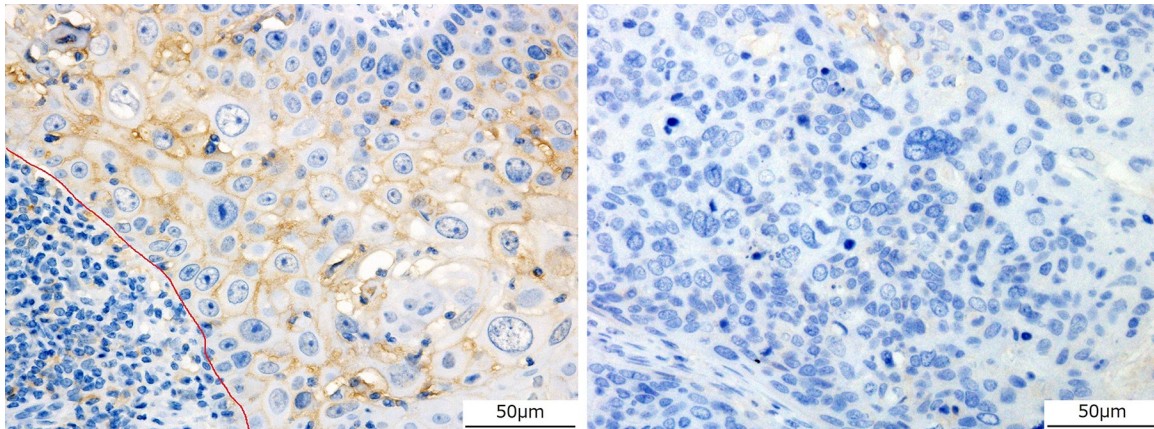

**Fig 2. Immunohistochemical analysis of PD-L1 (×40).** In the assessment of PD-L1 positivity (brown), the score was 1 when positive staining of the membrane was detected in >5% of the tumor cells (a), and 0 otherwise (b). The nuclei were counterstained with hematoxylin; red line shows the tumor border.

**Table 1. Clinical data of patients with cutaneous squamous cell carcinomas without Lymph Node metastasis (cSCC-).**

| Age (years) | Sex | Size (mm) | Tumor depth (mm) | Relapse | Prognosis (months) | Position | | | TILs | PD-L1 | Gene variation number |
|---|---|---|---|---|---|---|---|---|---|---|---|
| | | | | | | Face | Dorsal hand | Vulva | | | |
| 93 | F | 22 | 2 | - | 3 months, alive | 1 | | | 1 | 1 | 40 |
| 88 | M | 40 | 21 | - | 17 months, alive | 1 | | | 1 | 1 | 31 |
| 85 | M | 12 | 1 | - | 6 months, alive | 1 | | | 0 | 0 | 30 |
| 84 | M | 9 | 2 | - | 1 month, alive | 1 | | | 1 | 1 | 26 |
| 88 | F | 10 | 6 | - | 26 months, alive | 1 | | | 1 | 0 | 26 |
| 98 | F | 20 | 2 | - | 12 months, alive | 1 | | | 1 | 0 | 26 |
| 90 | M | 25 | 9 | - | 2 months, alive | 1 | | | 1 | 1 | 24 |
| 89 | M | 9 | 1 | - | 36 months, alive | 1 | | | 0 | 0 | 23 |
| 91 | F | 50 | 9 | - | 2 months, alive | 1 | | | 1 | 1 | 3 |
| 81 | F | 20 | 8 | skin | 27 months, alive | | 1 | | 1 | 0 | 113 |
| 82 | F | 15 | 15 | skin | 24 months, alive | | 1 | | 1 | 1 | 25 |
| 82 | F | 15 | 3 | - | 2 months, alive | | 1 | | 1 | 0 | 25 |
| 85 | M | 20 | 1 | - | 12 months, alive | | 1 | | 1 | 1 | 18 |
| 85 | F | 35 | 14 | - | 24 months, alive | | | 1 | 1 | 0 | 113 |
| | | | | | | 9 | 4 | 1 | 12/2[a] | 7/7[b] | 37[c] |

TILs, tumor-infiltrating lymphocytes (scores: 0, <5%; 1, >5%; a, total of 1/0 scores); PD-L1 (scores: 0, <5%; 1, >5%, b, total of 1/0 scores); c, average of the variation numbers.

## Results

The clinical summary, immunohistochemical results, and gene variation numbers (GVNs) are shown in Table 1 (cSCC without metastasis; cSCC-) and Table 2 (cSCC with metastasis; cSCC +). No therapies were administered to the patients before surgery and no patient had an immunosuppressive status. For the cSCC- and cSCC+ cases, respectively, the average ages were 87.2 and 83.0 years, average tumor sizes were 21.5 mm (range 9–50 mm) and 40.3 mm (range 4–90 mm), and tumor depths were 6.7 mm (range 2–21 mm) and 15.7 mm (range 1–60 mm). The average age was higher, and the tumor size was smaller in cSCC- patients. The tumor depth was higher in cSCC+ patients, and the tumor size was significantly larger in cSCC+ patients ($p < 0.05$). Cancer developed significantly in sun-exposed areas in 20 cases (face in 12 cases, dorsal hands in 5 cases, scalp in 1 case, and lips in 2 cases), while in 7 cases, cancer developed in non-sun-exposed areas (axillae in 2 cases, vulvae in 3 cases, thigh in 1 case, and foot in 1 case) ($p < 0.05$). In cSCC- patients, cancer developed predominantly in sun-exposed areas. The TIL score of cSCC- patients was higher than that of cSCC+ patients, and PD-L1 positivity tended to be lower in cSCC+ patients. A high TIL score was significantly related to PD-L1 positivity in cSCC- patients ($p < 0.05$).

The analytical details of gene sequencing are shown in Tables 3 and 4. The average passing filter (% PF), ≥% Q30 (Read 1), and >% Q30 (Read 2) were 57.95, 90.31, and 87.26%, respectively. The mean coverage was 1939.55 (93.85%). cSCC- cases had a total of 330 gene alterations, whereas cSCC+ cases had a total of 1333 gene alterations, excluding pathogenic benign or synonymous mutations. GVNs were higher in cSCC+ than in cSCC- cases. The majority of cSCC- cases occurred in sun-exposed areas, and the total GVNs were notably low. The lowest GVNs were in *RET*, *FGFR3* (synonymous), and *KDR* (not reported in ClinVar). In cSCC + cases, gene alterations were numerous and detected in all 50 investigated genes; however, alterations in *CTNNB1*, *GNAQ*, *GNAS*, *IDH2*, *JAK2*, and *NRAS* were not detected in cSCC-

**Table 2. Clinical data of patients with cutaneous squamous cell carcinomas with Lymph Node metastasis (cSCC+).**

| Age (years) | Sex | Size (mm) | Tumor depth (mm) | Relapse | Prognosis (months) | Position | | | | | | | | | TILs | PD-L1 | Gene variation number |
| --- | --- | --- | --- | --- | --- | --- | --- | --- | --- | --- | --- | --- | --- | --- | --- | --- | --- |
| | | | | | | Face | Lip | Scalp | Dorsal hand | Axilla | Buttock | Vulva | Thigh | Foot | | | |
| 81 | M | 25 | 5 | ND | ND | 1 | | | | | | | | | 1 | 1 | 716 |
| 82 | F | 4 | 1 | skin | 7 months, alive | 1 | | | | | | | | | 0 | 0 | 31 |
| 88 | F | 10 | 6 | - | 12 months, alive | 1 | | | | | | | | | 1 | 0 | 20 |
| 99 | F | 35 | 8 | skin | 12 months, alive | | 1 | | | | | | | | 0 | 0 | 236 |
| 85 | F | 28 | 7 | skin, bone | 72 months, alive | | 1 | | | | | | | | 1 | 1 | 28 |
| 72 | F | 50 | 10 | - | 33 months, alive | | | 1 | | | | | | | 1 | 1 | 141 |
| 82 | F | 15 | 4 | ND | ND | | | | 1 | | | | | | 0 | 0 | 368 |
| 79 | F | 90 | 60 | - | 6 months, alive | | | | | 1 | | | | | 0 | 1 | 123 |
| 91 | M | 45 | 2 | - | 41 months, alive | | | | | 1 | | | | | 1 | 0 | 27 |
| 60 | M | 60 | 24 | skin, LN | 3 months, death | | | | | | 1 | | | | 0 | 0 | 157 |
| 86 | F | 30 | 14 | - | 12 months, alive | | | | | | 1 | | | | 0 | 0 | 351 |
| 83 | M | 43 | 21 | skin | 20 months, alive | | | | | | | | 1 | | 0 | 0 | 41 |
| 92 | M | 90 | 43 | ND | ND | | | | | | | | | 1 | 1 | 0 | 82 |
| | | | | | | 3 | 2 | 1 | 1 | 2 | 2 | | 1 | 1 | 6/7[a] | 4/9[b] | 178[c] |

ND, no data; LN, lymph node; TILs, tumor-infiltrating lymphocytes (scores: 0, <5%; 1, >5%; a, total of 1/0 scores); PD-L1 (scores: 0, <5%; 1 >5%; b, total of 1/0 scores; c, average of the variation numbers.

cases. *ERBB4* and *NPM1* alterations were equally frequent, whereas *APC, ATM, CDKN2A, EGFR, ERBB2, FGFR2, FGFR3, MET, MPL, NOTCH1, PDGFRA, RB1, RET, SMAD4, SMARCB1, SMO,* and *TP53* alterations were more frequent in cSCC+ than in cSCC- cases. A summary of the newly detected gene alterations is presented in Table 4. Although these genes have not been reported to be pathogenic or malignant in ClinVar and PolyPhen, we detected more than 30% of cSCC cases with these alterations. There were some differences in the variant frequencies of specific genes, such as *ERBB4* (rs772717270) and *NPM1* (rs760834615), and *STK11* (rs2075606), between cSCC- and cSCC+ cases.

## Discussion

The results of our study indicate the relationship between the immunohistochemical reactivity of CD8/PD-L1 and gene alterations in cSCC with or without metastasis. Tumor metastasis was more common in cases wherein cancer did not arise in sun-exposed areas and was further related to TILs. Some biomarkers, such as PD-L1, TILs, microsatellite instability, mismatch repair, and tumor mutational burden (TMB), have been established for the prediction of immunotherapy effects in patients with metastatic disease [10]. In the current study, TIL numbers were higher in cSCC- than in cSCC+ cases, and they may suppress tumor metastasis. GVNs were decreased in the sun-exposed area because of the relationship between the sun-exposed area and the tumorigenesis of sun-related tumors [11]. Regardless of the cSCC- or cSCC+ status, sunlight might be associated with cSCC prognosis [11]. Although TMB was

**Table 3. Tumor gene mutation burden of squamous cell carcinomas.**

|  | cSCC+ | cSCC- |
|---|---|---|
| *ABL1* | 20 | 4 |
| *AKT1* | 14 | 1 |
| *ALK* | 12 | 3 |
| *APC* | 23 | 1 |
| *ATM* | 47 | 8 |
| *BRAF* | 6 | 2 |
| *CDH1* | 16 | 1 |
| *CDKN2A* | 40 | 3 |
| *CSF1R* | 26 | 16 |
| *CTNNB1* | 3 | 0 |
| *EGFR* | 60 | 6 |
| *ERBB2* | 25 | 2 |
| ***ERBB4*** | **66** | **55** |
| *EZH2* | 9 | 2 |
| *FBXW7* | 25 | 9 |
| *FGFR1* | 22 | 9 |
| *FGFR2* | 26 | 2 |
| *FGFR3* | 42 | 8 |
| *FLT3* | 27 | 14 |
| *GNA11* | 9 | 2 |
| *GNAQ* | 9 | 0 |
| *GNAS* | 12 | 0 |
| *HNF1A* | 10 | 3 |
| *HRAS* | 9 | 2 |
| *IDH1* | 5 | 2 |
| *IDH2* | 8 | 0 |
| *JAK2* | 11 | 0 |
| *JAK3* | 34 | 8 |
| *KDR* | 78 | 32 |
| *KIT* | 42 | 6 |
| *KRAS* | 17 | 4 |
| *MET* | 25 | 1 |
| *MLH1* | 4 | 2 |
| *MPL* | 14 | 1 |
| *NOTCH1* | 28 | 1 |
| ***NPM1*** | **41** | **40** |
| *NRAS* | 9 | 0 |
| *PDGFRA* | 18 | 1 |
| *PIK3CA* | 43 | 7 |
| *PTEN* | 42 | 10 |
| *PTPN11* | 12 | 4 |
| *RB1* | 47 | 6 |
| *RET* | 43 | 4 |
| *SMAD4* | 47 | 3 |
| *SMARCB1* | 26 | 2 |
| *SMO* | 20 | 3 |
| *SRC* | 13 | 1 |

(*Continued*)

**Table 3.** (Continued)

|  | cSCC+ | cSCC- |
|---|---|---|
| *STK11* | 45 | 16 |
| *TP53* | 74 | 17 |
| *VHL* | 29 | 6 |
| Benign/synonymous | 988 | 193 |
| Total | 2321 | 523 |

The genes in **BOLD** are those that have not been reported in cSCC, the genes in *BLUE* represent those with a higher frequency in cSCC+, and the genes in *RED* represent those detected only in cSCC+.

cSCC, cutaneous squamous cell carcinoma; cSCC-, cutaneous squamous cell carcinoma without metastasis; cSCC+, cutaneous squamous cell carcinoma with metastasis.

correlated with TILs, TILs were not related to GVN in this study [12]. In an immunohistological analysis, we set the PD-L1 cutoff to 5%, which did not seem to be related to tumor metastasis. Moreover, GVNs were not related to TILs/CD8 and PD-L1 positivity. However, the presence of PD-L1 was reported to be associated with poorer outcomes in metastatic/perineural cSCC [13, 14].

A number of reports on NGS of cSCC have been published [2, 5, 11, 15–20]. We detected alterations in *CSF1R*, *ERBB4*, *FGFR1*, *FLT3*, *KDR*, *STK11*, *PTEN*, and *NPM1*; among these, *ERBB4*, *FGFR1*, *STK11*, and *PTEN* have been highlighted in previous reports. *CSF1R*, *FGFR1*, *FLT3*, and *KDR* are related to the tumor microenvironment, and *ERBB4*, *STK11*, *PTEN*, and *NPM1* are related to proliferation-related factors [21–25]. These genetic mutations (due to

**Table 4. Summary of new gene alterations of cSCC.**

| cSCC- | cSCC+ | Gene | Chr | Coordinate | Type | Variant | Consequence | HGVSc | dbSNP ID |
|---|---|---|---|---|---|---|---|---|---|
| 9 (64%) | 9 (69%) | CSF1R | 5 | 149433596 | mnp | TG>TG/GA | downstream_gene_variant |  | rs386693509 |
| 11 (79%) | 11 (85%) | ERBB4 | 2 | 212578379 | insertion | T>T/TA | splice_region_variant, intron_variant | NM_005235.2:c.884-7dupT | rs769292151 |
| 10 (71%) | 13 (100%) | ERBB4 | 2 | 212578379 | deletion | TA>TA/T | splice_region_variant, intron_variant | NM_005235.2:c.884-7delT | rs397987661; rs67894136 |
| 9 (64%) | 12 (92%) | ERBB4 | 2 | 212578379 | deletion | TAA>TAA/T | splice_region_variant, intron_variant | NM_005235.2:c.884-8_884-7delTT | rs748883732 |
| 7 (50%) | 10 (77%) | ERBB4 | 2 | 212812097 | snv | T>C/C | intron_variant | NM_005235.2:c.421+58A>G | rs839541 |
| 2 (14%) | 6 (46%) | ERBB4 | 2 | 212578379 | deletion | TAAA>TAAA/T | splice_region_variant, intron_variant | NM_005235.2:c.884-9_884-7delTTT | rs772717270 |
| 5 (36%) | 5 (38%) | FGFR1 | 8 | 38285913 | deletion | GTCA>GTCA/G | inframe_deletion | NM_001174067.1:c.495_497delTGA | rs138489552 |
| 12 (86%) | 12 (92%) | FLT3 | 13 | 28610183 | snv | A>A/G, G/G | splice_region_variant, intron_variant | NM_004119.2:c.1310-3T>C | rs2491231 |
| 13 (93%) | 13 (100%) | NPM1 | 5 | 170837513 | deletion | CT>CT/C | intron_variant | NM_002520.6:c.847-5delT | rs34323200; rs397792554 |
| 10 (71%) | 11 (85%) | NPM1 | 5 | 170837513 | deletion | CTT>CTT/C | intron_variant | NM_002520.6:c.847-6_847-5delTT | rs766749752 |
| 9 (64%) | 11 (85%) | NPM1 | 5 | 170837513 | insertion | C>C/CT | intron_variant | NM_002520.6:c.847-5dupT | rs760834615 |
| 4 (29%) | 5 (38%) | PTEN | 10 | 89711833 | deletion | AT>AT/A | intron_variant | NM_000314.4:c.493-34delT |  |
| 11 (79%) | 13 (100%) | STK11 | 19 | 1220321 | snv | T>C/C, T/C | intron_variant | NM_000455.4:c.465-51T>C | rs2075606 |

HGVSc, Human Genome Variation Society (HGVS) notation in cDNA; dbSNP, The Single Nucleotide Polymorphism Database

chronic inflammation or growth durations, for instance) might lead to tumor invasion and metastasis of cSCC. The alterations in *CSF1R*, *FLT3*, and *NPM1* were detected in cSCC for the first time in this study. Comparing cSCC- and cSCC+, abnormalities in *CTNNB1*, *GNAQ*, *GNAS*, *IDH2*, *JAK2*, and *NRAS* were detected only in cSCC+; abnormalities in *APC*, *ATM*, *CDKN2A*, *EGFR*, *ERBB2*, *FGFR2*, *FGFR3*, *MET*, *MPL*, *NOTCH1*, *PDGFRA*, *RB1*, *RET*, *SMAD4*, *SMARCB1*, *SMO*, and *TP53* were more frequently detected in cSCC+ than in cSCC-. Of these genes, *GNAQ*, *GNAS*, and *JAK2* are associated with cell growth-related factors, whereas *NRAS*, *IDH2*, and *CTNNB1* are cancer-related genes [26–34]. *APC*, *PDGFRA*, *RET*, and *SMO* are tumor-related genes; *ATM*, *CDKN2A*, *RB1*, *NOTCH1*, *SMAD4*, *SMARCB1*, and *TP53* are tumor suppressor genes; *EGFR*, *ERBB2*, *MET*, and *MPL* are proliferation-related genes; and *FGFR2* and *FGFR3* are related to the tumor microenvironment. Thus, these genes seem to be related to tumor metastasis or malignancy [21, 22, 28–34]. From a genetic point of view, if these genes show some type of mutations, careful follow-up might be required after the primary diagnosis of cSCC without tumor metastasis. *ERBB4* and *NPM1* have similar mutation burdens in both cSCC+ and cSCC-; therefore, these genes are thought to be related to the tumorigenesis of cSCC.

Furthermore, we examined the relationship between tumor depth (>10 mm) and gene mutations to evaluate tumor invasion. Eight cases with tumor depth >10 mm showed mutations in *CSF1R*, *ERBB4*, *FLT3*, *KDR*, and *NPM1*, regardless of lymph node metastasis. As previously mentioned, these genes are related to the tumor microenvironment or tumor proliferation-related factors, and tumors with these gene alterations tend to have a strong invasion capacity and increased malignancy.

Although we present new findings in this report, there are a few limitations. First, the number of cases examined was small, and only a limited number of genes were searched. It has not yet been determined whether the detected mutations are tumorigenic. Progress from the early stages of tumor development and tumor growth cannot be observed during the clinical course. For CD8 and PD-L1, we assessed the overall area of the largest represented specimen, and evaluated the area with the highest number of CD8-/PD-L1-positive cells. Thus, the area with the highest number of CD8-/PD-L1-positive cells was different in each case. These results were only detected in cSCC, and we could not compare them with normal tissues because the size of the normal tissue was small for genetic analysis.

In conclusion, we demonstrated the relationship between CD8 and PD-L1 immunohistochemical reactivity and gene alterations in cSCC with or without metastasis. Tumor metastasis was more frequent in cases in which the cSCC occurred in a non-sun-exposed area, which was related to TILs or tumorigenesis of the cSCC in this study. Immunohistochemical positivity for PD-L1 did not seem to be related to the presence of tumor metastasis. From a genetic perspective, *ERBB4* and *NPM1* are assumed to be involved in tumorigenesis in cSCC. This may lead to new therapeutic applications and new drug development. Additionally, *GNAQ*, *GNAS*, *JAK2*, *NRAS*, *IDH2*, and *CTNNB1* are cancer-related genes that may be related to tumor metastasis. The genes that were highly expressed in metastatic cSCC (*APC*, *ATM*, *CDKN2A*, *EGFR*, *ERBB2*, *FGFR2*, *FGFR3*, *MET*, *MPL*, *NOTCH1*, *PDGFRA*, *RB1*, *RET*, *SMAD4*, *SMARCB1*, *SMO*, and *TP53*) may cause metastasis. If these genes contain mutations, careful follow-up may be required to prevent metastasis in cSCC.

## Author Contributions

**Conceptualization:** Haruto Nishida.

**Data curation:** Haruto Nishida, Yoshihiko Kondo, Takahiro Kusaba, Kazuhiro Kawamura, Yuzo Oyama.

**Formal analysis:** Haruto Nishida.

**Funding acquisition:** Haruto Nishida.

**Investigation:** Haruto Nishida.

**Methodology:** Haruto Nishida.

**Project administration:** Haruto Nishida, Tsutomu Daa.

**Resources:** Haruto Nishida.

**Supervision:** Haruto Nishida, Tsutomu Daa.

**Visualization:** Haruto Nishida.

**Writing – original draft:** Haruto Nishida.

**Writing – review & editing:** Haruto Nishida, Yoshihiko Kondo, Takahiro Kusaba, Kazuhiro Kawamura, Yuzo Oyama, Tsutomu Daa.

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
