## [Decision Letter · Decision Letter 0]

19 Dec 2022

PONE-D-22-26166The relationship between CD8/PD-L1 immunohistochemical reactivity and gene alterations in cutaneous squamous cell carcinomaPLOS ONE

Dear Dr. Nishida,

Thank you for submitting your manuscript to PLOS ONE. After careful consideration, we feel that it has merit but does not fully meet PLOS ONE’s publication criteria as it currently stands. Therefore, we invite you to submit a revised version of the manuscript that addresses the points raised during the review process.

We look forward to receiving your revised manuscript.

Kind regards,

Avaniyapuram Kannan Murugan, M.Phil., Ph.D.

Academic Editor

PLOS ONE

Journal Requirements

a) Did participants provide their written or verbal informed consent to participate in this study?

Grants: This study was supported by the KAKENHI grant from the Japan Society for the Promotion of Science KAKENHI Grant (Number 20K16195).

Additional Editor Comments:

Although manuscript received mostly positive comments, reviewers also raise many critiques about inadequate information on data, legend and accuracy in reference. Kindly address the critiques carefully in a point-by-point manner.

Reviewers' comments:

Reviewer's Responses to Questions

**Comments to the Author**

1. Is the manuscript technically sound, and do the data support the conclusions?

Reviewer #1: Partly

Reviewer #2: Yes

Reviewer #3: Partly

2. Has the statistical analysis been performed appropriately and rigorously? 

Reviewer #1: Yes

Reviewer #2: Yes

Reviewer #3: Yes

3. Have the authors made all data underlying the findings in their manuscript fully available?

Reviewer #1: No

Reviewer #2: Yes

Reviewer #3: No

4. Is the manuscript presented in an intelligible fashion and written in standard English?

Reviewer #1: Yes

Reviewer #2: Yes

Reviewer #3: Yes

5. Review Comments to the Author

Reviewer #1: The manuscript by Haruto Nishida et al., is a retrospective study that examined the CD8+ tumor infiltrating lymphocytes and and PD-L1 positivity in 27 cutaneous squamous cell carcinoma tissues. The authors also applied next-generation sequencing using a sequencing panel targeting 50 cancer-associated genes to examine the variation of these genes, and mutations in 4 novel genes were found. The results are primary and provide novel information on immune response and gene variation in the cutaneous squamous cell carcinoma, even though the number of cases is small.

The major drawback of the manuscript is in the result part. The specificity of the antibodies and the method of quantification should be clearly stated/presented, but was mostly missing in the manuscript. The only two figures included in the manuscript are not illustrated adequately and the figure legends did not provide necessary information. The tables give information on patients and major findings, but the legends were very confusing, partly due to the misuse of punctuation. As the authors pointed out in the discussion part, the lack of normal tissue control is another drawback of the study.

1. Evaluation of TILs and PD-L1

It was indicated in line 73-74 that “the one focus of the largest represented specimen was scored as 0 or 1 in the tumor invasion area …”. Does this mean that CD8-positive cells were counted only in one area of a single section? Does this result represent the percentage of CD8-positive cells in the whole specimen?

The same question of evaluation of PD-L1.

2. Figure 1 and Figure 2:

It is necessary to indicate the border of tumor.

The CD8 staining is brown, I suppose, but the counterstaining was not described at all, neither in the “material and method” part nor in the figure legends.

It would also be nice to include a scale bar in the figures.

The inclusion of a higher-magnification image would also be appreciated

3. Legends for table 1 and 2

The legends are very confusing and difficult to understand. For example, “<5%; 0, >5%; 1,”, does it mean that “0, <5%; 1, >5%;”?

4. Discussion

In this study, the TIL score is higer in non-metastatic cSSC, and the TIL score is correlated to the PD-L1 positivity. It has been reported before that within the established tumor microenvironment, infiltrating CD8+ T cells often fail to clear tumors (Bottomley et al., 2019); and within metastatic cSSC, PD-1 was frequently found on CD8+ cellsand the presence of PD-L1 was associated with poorer outcomes (Slater and Googe, 2016; Linedale et al., 2017). Could these issues been discussed further more?

Reviewer #2: Nishida et al. present a very nice study evaluating the relationship between CD8/PD-L1 biomarker expression with cutaneous squamous cell carcinoma gene alterations. This work is important for identifying genes that can be used for future therapeutic targets and in which cases these genes are relevant. I only have some minor comments to improve the manuscript prior to publication:

- In line 24, the sentence would be more precise with “We performed immunohistochemical staining for CD8 and PD-L1, ...” to differentiate between the NGS and immunohistochemical staining methods.

- In the introduction, the first sentence describes that several new immunotherapy drugs are in development, but you provide only one reference that is then used again in the following sentence. I suggest adding a recent review by Wright et al. (PMID: 34282763) that describes multiple targets and the clinical trial status which will help to strengthen your claim.

- The citation used in line 49 is incorrect since this refers to the AmpliSeq document from Illumina. Please correct this.

In the methods and figure legends, there is no description of the type of blue nuclear stain that is used in Figure 1 and 2. Please add such a description.

- To ensure reproducibility, I suggest reporting the actual P values for each statistical test used when you refer to the result as statistically significant.

Reviewer #3: The authors investigated the level of PD-L1 and TILs in cutaneous squamous cell carcinoma patients with or without metastasis. The Immunohistochemistry approach was used to determine the role of PD-L1 expression and frequency of CD8-positive tumor infiltrating lymphocytes (TILs) in these patients. In addition, authors investigated the correlations between the clinical data and PD-L1/CD8 expressions. Moreover, Nishida et al. examined the alternations in some cancer-associated genes using next-generation sequencing to identify potential gene alterations in this cohort. Their data showed that PD-L1 expression was lower in SCC cases with metastasis and TILs are more abundant in SCC cases without metastasis. Others also reported similar finding using the same approach (Roper E, et al. 2017. PMID: 28666643, Amoils M, et al. 2019. PMID: 30012051, Gambichler T, et al. 2017. PMID: 28501937, Kamiya S, et al. 2018. PMID: 30411509). However, it was concluded that the PD-L1 positivity was not related to the present of tumor metastasis in the discussion section.

Nishida et al. also found alterations in some cancer-associated genes such as CSF1R, ERBB4, FGFR1, FLT3, KDR, STK11, PTEN, and NPM1. Among the panel of 50 cancer-associated genes, abnormalities in CSF1R, KDR, FLT3, and NPM1 genes were claimed to be detected in cSCC for the first time. The alterations in CTNNB1, GNAQ, GNAS, IDH2, JAK2, and NRAS genes were only reported in cSCC with metastasis, while abnormalities in APC, ATM, CDKN2A, EGFR, ERBB2, FGFR2, FGFR3, MET, MPL, NOTCH1, PDGFRA, RB1, RET, SMAD4, SMARCB1, SMO, and TP53 were mostly reported in cSCC with metastasis.

Major concern

Although it was claimed that relationship between PD-L1/CD8 immunohistochemistry reactivity to gene alterations was investigated in this manuscript, but authors didn’t show any result that directly investigate this relationship. It would be interesting to see correlation between individual gene alteration to PD-L1/CD8 immunohistochemistry reactivity. Based on current result, Nishida et al. used two different approaches and reported their findings based on each approach to clinical data. Therefore, the title might be misleading.

Other concerns:

1. The figure legends were missing.

2. Both figures only have one image. Please show representative images of cases with PD-L1 <5% and >5% in Figure 2 and also TILs <5% and >5% in Figure 1.

3. Nishida et al. reported alteration for CSF1R, KDR, FLT3, and NPM1 genes in cSCC for the first time, however the KDR was previously reported in another study by Chang D, et al. 2021 (PMID: 34272401).

4. Please check lines 189/188 of manuscript and confirm that Blue represents those with higher frequency in cSCC- or cSCC+?

6. PLOS authors have the option to publish the peer review history of their article (what does this mean?). If published, this will include your full peer review and any attached files.

Reviewer #1: No

Reviewer #2: No

Reviewer #3: No

---

## [Author Response · Author response to Decision Letter 0]

6 Jan 2023

Dear Editor: 

Thank you for your review of our manuscript and for the positive comments. Below is our point-by-point response addressing the journal requirements, and comments from the reviewers and Editor. The revisions are highlighted in yellow in the manuscript.

Journal Requirements

We have carefully perused all the journal requirements and ensured that the manuscript complies with them.

a) Did participants provide their written or verbal informed consent to participate in this study? b) If consent was verbal, please explain i) why written consent was not obtained, ii) how you documented participant consent, and iii) whether the ethics committees/IRB approved this consent procedure.

The study was retrospective and patients’ consent was obtained using an opt-out method. Participants could always refuse if they wanted to. The study adhered to the guidelines described in the Declaration of Helsinki and was approved by the institutional ethics committee and review board of Oita University, Japan (approval number: 2191). We have described the details in “Material and Methods.”

Please state what role the funders took in the study.

The funder had no role in this study. Accordingly, we have declared the following in the cover letter: “The funders had no role in study design, data collection and analysis, decision to publish, or preparation of the manuscript.”

Reviewer: 1

Comment: 1. Evaluation of TILs and PD-L1

It was indicated in line 73-74 that “the one focus of the largest represented specimen was scored as 0 or 1 in the tumor invasion area …”. Does this mean that CD8-positive cells were counted only in one area of a single section? Does this result represent the percentage of CD8-positive cells in the whole specimen? The same question of evaluation of PD-L1.

Response: Thank you for your query. For CD8 and PD-L1, we assessed the overall area of the largest represented specimen and evaluated the area with the highest number of CD8-/PD-L1-positive cells. Thus, the percentage of the highest area was different in each case. We have added this point as a limitation of the present study.

Comment: 2. Figure 1 and Figure 2: It is necessary to indicate the border of tumor. The CD8 staining is brown, I suppose, but the counterstaining was not described at all, neither in the “material and method” part nor in the figure legends. It would also be nice to include a scale bar in the figures. The inclusion of a higher-magnification image would also be appreciated

Response: Thank you for the valuable suggestions. We have added a scale bar and a red line indicating the tumor border in the figures. The nuclei were counterstained with hematoxylin. We have added the information in the “Material and Methods” and figure legends. The image is presented at ×40 (the highest magnification). 

Comment: 3. Legends for table 1 and 2: The legends are very confusing and difficult to understand. For example, “<5%; 0, >5%; 1,”, does it mean that “0, <5%; 1, >5%;”?

Response: We apologize for the confusing presentation. We have corrected it as per your suggestion.

Comment: 4. Discussion: In this study, the TIL score is higer in non-metastatic cSSC, and the TIL score is correlated to the PD-L1 positivity. It has been reported before that within the established tumor microenvironment, infiltrating CD8+ T cells often fail to clear tumors (Bottomley et al., 2019); and within metastatic cSCC, PD-1 was frequently found on CD8+ cells and the presence of PD-L1 was associated with poorer outcomes (Slater and Googe, 2016; Linedale et al., 2017). Could these issues been discussed further more?

Response: Thank you for your comment and for drawing our attention to these previously published reports. We carefully referred to these reports and have added the following text to the “Discussion” section: “the presence of PD-L1 was reported to be associated with poorer outcomes in metastatic/perineural cSCC [13, 14].” 

Reviewer: 2

Comment: In line 24, the sentence would be more precise with “We performed immunohistochemical staining for CD8 and PD-L1, ...” to differentiate between the NGS and immunohistochemical staining methods.

Response: Thank you for your comment. We have rephrased the sentence as suggested by you. 

Comment: In the introduction, the first sentence describes that several new immunotherapy drugs are in development, but you provide only one reference that is then used again in the following sentence. I suggest adding a recent review by Wright et al. (PMID: 34282763) that describes multiple targets and the clinical trial status which will help to strengthen your claim.

Response: Thank you for your suggestion. We agree with your comment and have referred to the review article mentioned by you.

Comment: The citation used in line 49 is incorrect since this refers to the AmpliSeq document from Illumina. Please correct this. In the methods and figure legends, there is no description of the type of blue nuclear stain that is used in Figure 1 and 2. Please add such a description. 

Response: Thank you for your pointing out these issues. We have corrected the “References.” The nuclei were counterstained with hematoxylin. We have added the information in the “Material and Methods” and figure legends.

Comment: To ensure reproducibility, I suggest reporting the actual P values for each statistical test used when you refer to the result as statistically significant.

Response: Thank you for your suggestion. We have added p < 0.05 wherever needed.

Reviewer: 3

Comment: Based on current result, Nishida et al. used two different approaches and reported their findings based on each approach to clinical data. Therefore, the title might be misleading.

Response: Thank you for your suggestion. We have changed the title to “CD8/PD-L1 immunohistochemical reactivity and gene alterations in cutaneous squamous cell carcinoma.” 

Comment: 1. The figure legends were missing. 

Response: Thank you for your comment. We have included the Figure legends in the body of the text immediately after the first paragraph in which the figures are cited, per the submission guidelines.

Comment: 2. Both figures only have one image. Please show representative images of cases with PD-L1 <5% and >5% in Figure 2 and also TILs <5% and >5% in Figure 1.

Response: Thank you for this suggestion. We have added images for cases with PD-L1 <5% and TILs <5%.

Comment: 3. Nishida et al. reported alteration for CSF1R, KDR, FLT3, and NPM1 genes in cSCC for the first time, however the KDR was previously reported in another study by Chang D, et al. 2021 (PMID: 34272401). 

Response: Thank you for this suggestion. We have incorporated the correction in the revised manuscript.

Comment: 4. Please check lines 189/188 of manuscript and confirm that Blue represents those with higher frequency in cSCC- or cSCC+.

Response: Thank you for this suggestion. We have corrected it as “cSCC+.”

---

## [Decision Letter · Decision Letter 1]

30 Jan 2023

CD8/PD-L1 immunohistochemical reactivity and gene alterations in cutaneous squamous cell carcinoma

PONE-D-22-26166R1

Dear Dr. NIshida,

We’re pleased to inform you that your manuscript has been judged scientifically suitable for publication and will be formally accepted for publication once it meets all outstanding technical requirements.

Kind regards,

Avaniyapuram Kannan Murugan, M.Phil., Ph.D.

Academic Editor

PLOS ONE

Additional Editor Comments (optional):

Reviewers' comments:

Reviewer's Responses to Questions

**Comments to the Author**

1. If the authors have adequately addressed your comments raised in a previous round of review and you feel that this manuscript is now acceptable for publication, you may indicate that here to bypass the “Comments to the Author” section, enter your conflict of interest statement in the “Confidential to Editor” section, and submit your "Accept" recommendation.

Reviewer #1: All comments have been addressed

Reviewer #2: All comments have been addressed

Reviewer #3: All comments have been addressed

2. Is the manuscript technically sound, and do the data support the conclusions?

Reviewer #1: Yes

Reviewer #2: Yes

Reviewer #3: (No Response)

3. Has the statistical analysis been performed appropriately and rigorously? 

Reviewer #1: Yes

Reviewer #2: Yes

Reviewer #3: (No Response)

4. Have the authors made all data underlying the findings in their manuscript fully available?

Reviewer #1: Yes

Reviewer #2: Yes

Reviewer #3: (No Response)

5. Is the manuscript presented in an intelligible fashion and written in standard English?

Reviewer #1: Yes

Reviewer #2: Yes

Reviewer #3: (No Response)

6. Review Comments to the Author

Reviewer #1: (No Response)

Reviewer #2: I thank the authors for their work in addressing my comments. I find that all of the revisions made have improved the paper and it is now acceptable for publication.

Reviewer #3: (No Response)

7. PLOS authors have the option to publish the peer review history of their article (what does this mean?). If published, this will include your full peer review and any attached files.

Reviewer #1: No

Reviewer #2: No

Reviewer #3: No

---

## [Editor Report · Acceptance letter]

1 Feb 2023

PONE-D-22-26166R1 

CD8/PD-L1 immunohistochemical reactivity and gene alterations in cutaneous squamous cell carcinoma 

Dear Dr. Nishida:

I'm pleased to inform you that your manuscript has been deemed suitable for publication in PLOS ONE. Congratulations! Your manuscript is now with our production department. 

Kind regards, 

on behalf of

Dr. Avaniyapuram Kannan Murugan 

Academic Editor

PLOS ONE